# Marital Relationship and Quality of Life in Couples Following Hip Replacement Surgery

**DOI:** 10.3390/life11050401

**Published:** 2021-04-28

**Authors:** Michael Tanzer, Christopher Pedneault, Esther Yakobov, Adam Hart, Michael Sullivan

**Affiliations:** 1Division of Orthopaedic Surgery, McGill University, Montréal, QC H3A 0G4, Canada; christopher.pedneault@mail.mcgill.ca (C.P.); adam.hart@mcgill.ca (A.H.); 2Department of Psychology, McGill University, Montréal, QC H3A 0G4, Canada; esther.yakobov@mail.mcgill.ca (E.Y.); michael.sullivan@mcgill.ca (M.S.)

**Keywords:** arthroplasty, hip, outcome, spouse, marriage, quality of life

## Abstract

For the majority of patients with osteoarthritis, total hip (THA) arthroplasty results in a significant reduction in pain, emotional distress, and disability and a significant improvement in their quality of life. Little is known about how these recovery-related changes impact the spouse or the marital relationship. Methods: Twenty-nine couples whose spouse underwent a THA (29 THA) participated in a semi-structured retrospective interview designed for this study. They were each asked to recall the level of pain before and after surgery and to provide a numerical rating score for questions pertaining to the level of disability in seven different activities of daily living. Couples were also asked to list in order of importance the five ways in which the surgery affected their overall quality of life. Results: The spouses estimated their partner’s pain, both pre- and postoperatively, to be significantly higher level than the patient’s perception. The spouses perceived a greater improvement in family/home responsibilities, recreation and social activities, and in their occupation than that noted by the partner. After the arthroplasty, the spouses indicated that their lives had improved with respect to doing more activities/leisure (72%), because their partner had less suffering (59%), they had more independence/less caregiving (55%), it improved their marital relationship (52%), they had a better social/family life (28%), and they were able to travel (28%). Conclusions: This study indicates that THA result in a significant improvement in quality of life not only for the patients, but also for their spouses.

## 1. Introduction

The pain of arthritic disorders occurs in a social and environmental context; therefore, the pain and disability of arthritis not only can affect the patient but also the spouse [1]. Individuals with persistent pain conditions such as osteoarthritis are at greater risk for the development of problematic health and mental health outcomes, such as heightened pain severity, disability, and depression [2]. These adverse health and mental health outcomes are not restricted to the individual with arthritis but are also likely to impact on the spouse [1].

Spouses of individuals with chronic illness most frequently assume the role of informal caregivers [3]. Informal caregivers tend to provide assistance with daily living and often reduce their working hours, assume more responsibilities with household chores, and make other adjustment due to the increased burden of caregiving. Numerous investigations have shown that the spouses of individuals with persistent pain report lower levels of marital satisfaction, higher rates of depression, and a lower quality of life [4,5].

For the vast majority of individuals with osteoarthritis, joint replacement surgery will result in significant reductions in pain, emotional distress, and disability and there will be a significant improvement in their function [6]. Little is currently known about how these recovery-related changes impact the spouse or the marital relationship. It is possible that when the patients’ health and mental health status improve, so does the health and mental health status of the spouse, and in turn, the quality of the relationship. However, it is also possible that long periods of spousal caregiving for a patient experiencing persistent pain, distress, and disability contribute to enduring spousal distress and marital dysfunction [7].

We undertook a study to evaluate the spouse’s perception of the patient’s pain and disability before and after hip replacement surgery and to assess the degree to which patients and their spouses perceive themselves to be disabled, both preoperatively and postoperatively. In addition, we determined the ways in which the spouse’s personal and marital life improved after the patient’s successful hip arthroplasty.

## 2. Methods

### 2.1. Participants

The study sample comprised 29 heterosexual couples (58 respondents). The average age of the patients was 68 years old (range, 48–86) and 67 years (range, 48–84) for the spouses. There were 14 male patients and 15 female patients, and 15 female spouses and 14 male spouses. All the 29 THAs were performed by the senior author (M.T.). Patients had been living with a diagnosis of osteoarthritis for an average of 7 years (range, 0.5–50), while the average waiting time to receive an operation was 8.9 months (range, 2–24). On average, couples had been married for 36.7 years (range, 6.0–62.5). The levels of education and number of children are also summarized in Table 1 with the aforementioned data. Five couples refused to be part of the study because they did not want to discuss their sexual behavior in their marriage.

### 2.2. Procedure

Hospital institutional review board approval (MUHC Authorization (2017–1658, MUHC-16-091/eReviews_5492) was obtained prior to the onset of the study. Hip arthroplasty patients were identified either from our arthroplasty database or were recruited for the study at their 6-week follow-up visit and consented to participate. All patients in the study were operated on by a single surgeon (M.T.), using the same implants and at a single institution. All patients in the study had to be 3–12 months following their surgery, had a diagnosis of osteoarthritis (OA), were married or co-habiting with their spouse for at least 5 years prior to surgery, and were between 65 and 85 years of age. Patients were excluded if they either received an arthroplasty for a diagnosis other than OA, were divorced, had other chronic pain or illnesses that significantly affected their ability to ambulate, had other lower extremity joints that required replacement, or had a serious postoperative complication that affected their recovery. Couples participated in a semi-structured interview, and the interview took place either in clinic or was conducted over the telephone. The interviews took place an average of 11.8 months after surgery (range, 1.8–29.7). In total, 4 couples were interviewed 3 months postoperatively, 6 couples at 6 months, 2 couples at 9 months, 5 couples at 12 months, and 12 couples over 1 year postoperatively. During the interview, couples were interviewed separately. This was done in the early postoperative period, so that the spouse and the patient could readily recall the pain and disability, but after the patient had recovered from surgery and was fully functional (Harris Hip Score of 100). The interview was done when the patient had a full recovery with no pain or limitations, indicated by their normal Harris Hip Score of 100. The patients ranged in age from 65 to 85, resulting in the time to obtain a normal Harris Hip Score being between 3 and 12 months, at which time the interview was done. Basic descriptive analysis for nominal data was used, and a paired Student’s T test was used to compare pre-operative and post-operative pain and PDI scores, and a *p* value of *p* < 0.05 was deemed statistically significant. The datasets used and/or analyzed during the current study are available from the corresponding author on reasonable request.

## 3. Measures

Interview: A semi-structured interview was developed for the purpose of this study. Interview questions were generated to address patients’ and spouses’ perceptions of patients’ pain severity, as well as pain-related disability preoperatively and postoperatively at the time of the survey. The spouses were also asked to provide demographic data for the duration of their marriage, level of education, and number of children. Finally, the spouses and the patients were asked to list in order of importance the five ways in which the surgery affected their overall quality of life.

Pain Severity: Patients and spouses were asked to rate the severity of the patient’s pain on a 0–10 numerical rating scale (NRS) with the endpoints (0) no pain, and (10) unbearable pain.

Self-Reported Disability: The Pain Disability Index (PDI) was used to measure the degree to which patients perceived themselves to be disabled by their pain, both preoperatively and postoperatively at the time of the survey, in 7 different areas of daily living (home, social, recreational, occupational, sexual, self-care, life support) (Figure 1) [8]. Items 1–5 of the PDI are considered voluntary activities, while items 6 and 7 are considered obligatory activities [9]. For each life domain, respondents were asked to provide perceived disability ratings on 10-point scales with the endpoints (0) *no disability* and (10) *total disability*. The PDI has been shown to be internally reliable and significantly correlated with objective indices of disability [10]. The rating scales were designed to measure how much the patient’s pain is preventing him/her from doing what he/she would normally do or from doing it as well as he/she normally would. The instructional set of the PDI was modified in order to assess how spouses perceived their partners’ level of disability. Both a one-factor (total score) and two-factor solution (voluntary item and obligatory item score) were calculated for the PDI score.

## 4. Results

All patients reported that they had completely recovered from their surgery at the time of the study. Overall, the patients rated the level of their preoperative pain significantly lower than did their spouse. The patients rated the level of their preoperative pain as 7.4 out of 10, while their spouses rated their preoperative pain as 8.3 (*p* = 0.05). Postoperatively, at the time of the survey, the patients felt their pain had improved to 0.9, while their spouses continued to report their pain as significantly worse, being 1.4 (*p* = 0.05).

PDI scores averaged 33.6 for the patients preoperatively (Table 2). The patients’ pain impacted their voluntary activities more than their obligatory activities. Overall, patients identified recreation as the most restricted activity. The preoperative PDI scores reported by the spouses were significantly greater than those reported by the patients. They also reported that the patient’s pain restricted his/her ability to be involved in recreation activities as the greatest disability. Of the seven different areas of daily living, only sexual behavior and self-care activities were scored lower by the spouses than the patients. Overall, the patients reported a significant reduction in disability (20.7 voluntary, 5.2 obligatory) (*p* < 0.001), and spouses reported a mean improvement of 28.7 points (23.6 voluntary, 5.1 obligatory) (*p* < 0.001).

For patients, the most common significant ways in which the surgery affected their quality of life, in order of importance, was the noticeable improvement in mobility (93%), the fact that they could resume their favorite leisure and sporting activities (76%), their remarkable improvement in pain (72%), and improvement in their social and family lives (38%) (Table 3). Ability to travel, more independence, and a general sense of well-being was also enjoyed by many patients. The spouses reported that the main advantage of the patients’ arthroplasty surgery was the ability to carry on with social and leisure activities with their partner (72%) (Table 4). The other benefits of arthroplasty endorsed by spouses, in order of importance, was that they no longer witnessed the patient suffering (59%), a diminished caregiver burden and a sense of independence (55%), an improved marital relationship (52%), an improved social and family life (28%), and the freedom to travel (28%).

## 5. Discussion

This survey study highlights the many benefits of hip arthroplasty surgery, not only for the patient, but also for their spouse. It also underlines the differences in perceived pain and levels of disability in relation to activities of daily living between both the patient and their spouse. When compared to patients, spouses reported higher levels of pain, both preoperatively and at the time of the survey. The spouses also perceived a greater overall reduction in pain-related disability across activities of daily living. Importantly, we have demonstrated that the subjective benefits of surgery extend to spouses of arthroplasty patients. After the surgery, the spouses indicated that their lives had improved with respect to partaking in physical and leisure activities with their spouse. They also perceived less partner suffering and had more independence with less caregiving, and consequently, the surgery improved their marital relationship.

The discrepancy between pain perceptions reported by patients and their spouses has been previously reported in several studies in a couples context [11,12,13,14]. In cancer patients, Clipp et al. noted that the lowest levels of agreement between patient and spouse-reported outcomes pertained to pain and coping with illness [12]. Other studies have also found that caregivers more often perceived patients having more pain than patients themselves [11,12,13,14]. They hypothesized that patient reports may be inflated or optimistic compared with more realistic caregiver ratings, or on the other hand, negative reports from the spouse may be associated with the significant burden of care and higher mental health issues experienced by caregivers [12]. For patients suffering from chronic musculoskeletal pain, Cano et al. came to similar conclusions, as spouses rated patients’ pain more severely [11]. They suggested that spouses may find verbalizations of pain difficult to avoid, which may heighten spouses’ distress and therefore their ratings of pain.

Although the exact etiology for the differences in pain perceptions and levels of disability remain unclear, this study has shown that there are many positive effects of hip arthroplasty for the spouse. When asked the five most important ways in which the surgery affected their lives specifically, the spouses responded positively, and recurring themes arose such as improved leisure activities, less caregiving, more independence, and less perceived suffering.

The results of this study indicate that the patient’s pain and function are most affected by their arthritis. This is in line with the present indications for joint replacement, primarily pain and loss of function. Preoperatively, patients are generally able to modify their lives to varying degrees in order to accommodate their arthritis. Therefore, it is reasonable to expect that the patients and their spouses in this study found the greatest benefit of the hip replacement was related to the pain reduction and increased mobility after the total hip replacement. This differs from more life threatening or chronic disease where life modification is drastic or not possible. In these cases, mood/well-being can be significantly affected as well.

The limitations of this study pertain to the survey nature of it. Consequently, it does not shed light on the particular causes for the disparities between the questionnaire responses of the patients and their spouse. In addition, as this study is retrospective in nature and only done after the arthroplasty, it may be limited by recall bias of both the patient and their spouse. In addition, the Harris Hip Score was only used in this study to include/exclude patients. It did not give any information on patient’s satisfaction or health-related quality of life (HRQoL). The inclusion of only patients with a Harris Hip Score of 100 may have excluded patients with severe comorbidities and postoperative complications, which can impact the generalizability of the study findings. Furthermore, the quality of the marital relationship and the psychological status of the couples were not determined, so it was not possible to determine whether or not these couples are representative of a normal clinical sample. Finally, the relatively small number of couples in this study may limit the conclusions of this study, as well as the generalizability of these findings to couples of differing socioeconomic status, ethnicity, and duration of marriage.

It has become clear that since its inception in the 1960s by Sir John Charnley, total hip arthroplasty has evolved into a reliable and reproducible surgical procedure to restore the patient’s function [15]. The aging population and the success of the procedure is now driving an increase in the number of THA done each year. This increase in the number of THA performed each year and the resultant increase in health-care expenditures has highlighted the need for consistent evaluation of the effectiveness of this surgery. Historically, the results of hip arthroplasty were assessed through measures of mortality and morbidity rates, operative complications, and the lifetime survivorship of the implants. However, improvements in the medical and surgical aspects of the procedure and implant design have improved significantly, such that these indicators alone are not sufficient to determine the health care benefits of THA. Other important outcomes, such as health and health-related outcomes, need to be factored into the outcome assessment. The evaluation of health-related quality of life through a validated and patient-completed questionnaire has become a standard approach for this. Numerous studies have demonstrated substantial improvements in the scores of the physical health of the patient following their THA [16,17,18,19,20,21,22,23,24,25,26,27,28,29,30]. Although the greatest improvement seemed to take place within the first three to six months after surgery, studies with longer-term follow-up have also found long-lasting improvement [31,32,33,34]. To date, the overall effectiveness of THA in terms of health-related quality of life has been directed at the patient. Previously, it has been shown that spouses of patients undergoing hip or knee replacement play an important role in the early recovery process [35]. This study specifically addresses the benefits of hip arthroplasty surgery for the patient’s spouse. The improvements in mobility and pain allow patients to partake in activities such as walking, travelling, and sporting activities with their spouse, which are reported in this study to have a positive effect on their quality of life.

## 6. Conclusions

In the era of patient-centered care, it should be recognized that a hip arthroplasty has the potential to improve the quality of life and marital relationship not only for the patient, but for the spouse as well.

## Figures and Tables

**Figure 1 life-11-00401-f001:**
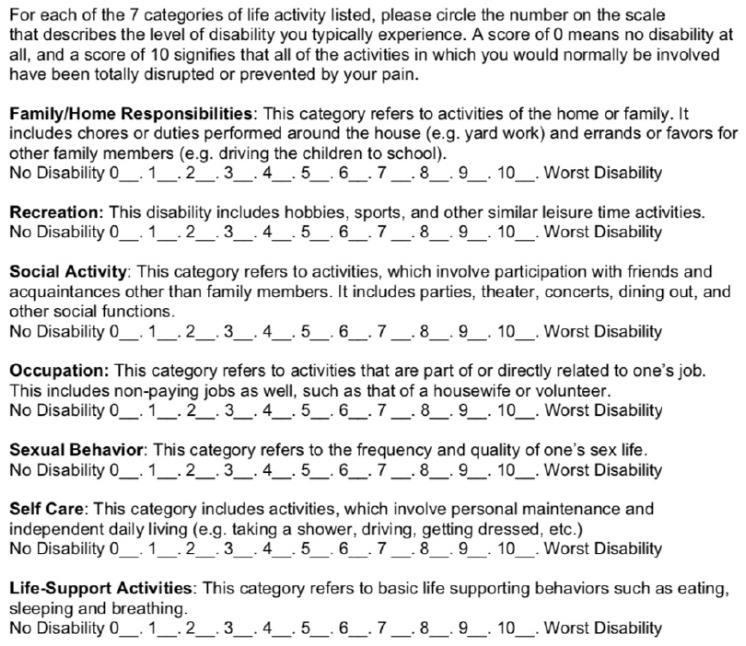
The Pain Disability Index, the patient-reported outcome measure on activities of daily living.

**Table 1 life-11-00401-t001:** Demographic data of patients and their spouses.

Demographics	Patient	Spouse
Mean Age (years)	68.4	67.3
Male	14	15
Female	15	14
High-school Education	11	12
University Education	15	13
Post-graduate Education	3	4
Years of marriage	36.7	
Mean Number of Children	2	
Mean Preopertive Waiting time (months)	8.9	
Mean years living with osteoarthritis	7	

**Table 2 life-11-00401-t002:** Patient and spouse-reported disability scores before and after arthroplasty surgery.

	Disability Score Pre-op (Mean)		Disability Score Post-op (Mean)		Improvement	
	Patient	Spouse	Patient	Spouse	Patient	Spouse
**Family/Home responsibilities**	5.7	6.2	1.4	0.9	4.3	5.3
**Recreation**	7.0	7.2	1.8	1.1	5.2	6.1
**Social Activity**	5.0	5.4	0.8	0.8	4.2	4.4
**Occupation**	5.2	5.5	1.4	1.0	3.8	4.5
**Sexual behavior**	4.9	4.0	1.7	0.8	3.2	3.2
**Self-care**	3.6	3.0	0.5	0.6	3.1	2.4
**Life-support activity**	2.2	3.2	0.1	0.4	2.1	2.8
**Total**	33.6	34.5	7.7	5.6	25.9	28.7

**Table 3 life-11-00401-t003:** Patient-reported improvements in quality of life.

Quality of Life Measures	Patients (Number of Respondents)
Mobility	27/29
Leisure and sporting activities	22/29
Less pain	21/29
Social/family life	11/29
Travelling	7/29
Mood/Well-being	6/29
Independence	5/29
Work	4/29
Other	6/29

**Table 4 life-11-00401-t004:** Spouse-reported improvements in quality of life.

Quality of Life Measures	Spouse (Number of Respondents)
Activities/leisure	21/29
Less partner suffering	17/29
More independence/less caregiving	16/29
Improved relationship	15/29
Social/family life	8/29
Travelling	8 /29
Others	5/29

## Data Availability

Not applicable.

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
