# Peer review of "Marital Relationship and Quality of Life in Couples Following Hip Replacement Surgery"

_life, 2021, doi:10.3390/life11050401_

Round 1

Reviewer 1 Report

This is a well-written manuscript about the marital relationship between partners in THA. Originality is very high. Super idea for publication. Congratulation!

Reading it some questions arose to me that should be clarified.

Title: Since only total hips were followed this should be indicated in the title. Otherwise, it is misleading.

Abstract: One of the weak points of the study is the retrospective nature of the interview. This should be mentioned in the Abstract methods.

2.2. The number of the approval of the review board should be mentioned.

How many patients had to be excluded from the study?

Why did you choose the 6-week follow-up visit to recruit? Why didn’t you do a prospective study (which really seems to be a no go to these kind of studies).

Why did you choose 5 different timepoints for the interview? Did this influence the quality and result of the answers? i.e. were the answers different dependent on time passed after the surgery?

I guess the HHS was only used to include/exclude patients. It does not give any information on patient’s satisfaction or health-related quality of life (HRQoL). May you could point that out.

  1. Limitation besides the retrospective nature of the study and the small number of participants is that you divided them further into different timepoints. I don’t know, if you can draw the right conclusions with such small numbers…

LL 205-206 you mention a validated and patient-completed questionnaire. Is that what you really did?

Author Response

Responses to Reviewers

I would like to than the reviewers for their expertise and queries. The manuscript has been modified to respond to and/or clarify the issues raised by the reviewers

Title: Since only total hips were followed this should be indicated in the title. Otherwise, it is misleading.

We agree with the reviewer and the title has been changed to:

Marital Relationship and Quality of Life in Couples Following Hip Replacement Surgery

Abstract: One of the weak points of the study is the retrospective nature of the interview. This should be mentioned in the Abstract methods.

We agree with the reviewer and the abstract has been modified to:

Twenty-nine couples whose spouse underwent a THA (29 THA) participated in a semi-structured retrospective interview designed for this study.

2.2. The number of the approval of the review board should be mentioned.

The IRB approval - MUHC Authorization (2017-1658, MUHC-16-091 / eReviews_5492) is now indicated in the manuscript:

Hospital institutional review board approval (MUHC Authorization (2017-1658, MUHC-16-091 / eReviews_5492)was obtained prior to the onset of the study.

How many patients had to be excluded from the study?

This is now clarified in the manuscript on lines 68-70: “Five couples refused to be part of the study because they did not want to discuss their sexual behaviour in their marriage.”

Why did you choose the 6-week follow-up visit to recruit? Why didn’t you do a prospective study (which really seems to be a no go to these kind of studies).

Thank you for your comment. As pointed out, a prospective study can result in some bias by the participants because they could be actively looking for the changes in their lives or discussing the changes before and after surgery. Although the retrospective nature of the study is not ideal, it was done close enough to the time of the surgery to help minimize the loss of remembering the issues questioned both pre and postoperatively.

The 6-week visit was the first routine visit that the patients were seen postoperatively. This allowed us to recruit patients shortly after surgery, so that once achieved full function, at 3-12 months, they would already be part of the study and could be interviewed while still having a good recollection of the preoperative issues and appreciate the postoperative benefits. As noted in lines 88-91 “This was done in the early postoperative period, so that the spouse and the patient could readily recall the pain and disability, but after the patient had recovered from surgery and was fully functional (Harris Hip Score of 100).”

Why did you choose 5 different timepoints for the interview? Did this influence the quality and result of the answers? i.e. were the answers different dependent on time passed after the surgery?

As noted in the manuscript on lines 88-91:” This was done in the early postoperative period, so that the spouse and the patient could readily recall the pain and disability, but after the patient had recovered from surgery and was fully functional (Harris Hip Score of 100).” The interview was done when the patient had full recovery and had no pain or limitations according to the Harris Hip Score. As noted in the manuscript, the patients ranged in age from 65-85 so that their recuperation to a normal Harris Hip Score was somewhat variable between 3-12 months.

To clarify this further to the reader, the following sentence has been added: “The interview was done when the patient had a full recovery with no pain or limitations, indicated by their normal Harris Hip Score of 100. The patients ranged in age from 65-85, resulting in the time to obtain a normal Harris Hip Score being between 3-12 months, at which time the interview was done”.

I guess the HHS was only used to include/exclude patients. It does not give any information on patient’s satisfaction or health-related quality of life (HRQoL). May you could point that out.

The reviewer is correct. This limitation is now noted in the limitations section: “In addition, the Harris Hip Score was only used in this study to include/exclude patients. It does not give any information on patient’s satisfaction or health-related quality of life (HRQoL).

Limitation besides the retrospective nature of the study and the small number of participants is that you divided them further into different timepoints. I don’t know, if you can draw the right conclusions with such small numbers…

Yes this is a potential issue. The limitation paragraph in the Discussion has been modified to read: “Finally, the relatively small number of couples in this study may limit the conclusions of this study, as well as the generalizability of these findings to couples of differing socio-economic status, ethnicity and duration of marriage.”

LL 205-206 you mention a validated and patient-completed questionnaire. Is that what you really did?

Certainly the questionnaires used in this study, including the HHS, are all validated in the literature.

Reviewer 2 Report

It is a new way to look on life quality in patients after THA

Clear description of the material and methods

Adequate methods used to evaluate the outcome and be able to answer the research question

Results well presented

Good language

A limitation is the small number of patients

However overall an interesting study.

Author Response

Responses to Reviewers

I would like to than the reviewers for their expertise and queries. The manuscript has been modified to respond to and/or clarify the issues raised by the reviewers

It is a new way to look on life quality in patients after THA

Clear description of the material and methods

Adequate methods used to evaluate the outcome and be able to answer the research question

Results well presented

Good language

A limitation is the small number of patients

We agree that the number of patients is small. However, the results are consistent with all the couples. None-the-less, we have modified the limitation paragraph in the Discussion section to: “Finally, the relatively small number of couples in this study may limit the conclusions of this study, as well as the generalizability of these findings to couples of differing socio-economic status, ethnicity and duration of marriage.”

However overall an interesting study.

Reviewer 3 Report

I have had the opportunity to review the important and interesting contribution entitled "Marital Relationship and Quality of Life in Couples Following Joint Replacement Surgery "

Authors present a survey involving not only pateints but also their spouses and find that the perception of pain /disability is different, depending on the perspective as patient or as companion. As the tables 3 & 4 reveal, change in some domains (eg mood/well-being) affects only a small proportion of patients, whereas in others the relief is there for almost all patients.  What explanations do the authors offer for such disparities?  The prevalence of complicating factors? In this context: Was the quality of the patient/spouse relationship assessed in any manner? Furthermore, I recommend the following paper : PMID 27939622.  It informs well about the subjective view of OA on the patients´ part (and doctors´, too).

That aside, I was wondering about the inclusion criteria: is the study sample representative of a normal clinical sample? Were they screened for any psychiatric disorder? What role did socioeconomic aspects play?

Would the spouses have to be screened too for any co-morbidity?

Author Response

Responses to Reviewer 3

I would like to than the reviewers for their expertise and queries. The manuscript has been modified to respond to and/or clarify the issues raised by the reviewers

Authors present a survey involving not only patients but also their spouses and find that the perception of pain /disability is different, depending on the perspective as patient or as companion.

As the tables 3 & 4 reveal, change in some domains (eg mood/well-being) affects only a small proportion of patients, whereas in others the relief is there for almost all patients.  What explanations do the authors offer for such disparities?  The prevalence of complicating factors? In this context: Was the quality of the patient/spouse relationship assessed in any manner?

The Reviewer brings up a very important issue. I have clarified this further by adding the following paragraph in the Discussion section:

“The results of this study indicate that the patient’s pain and function are most affected by their arthritis. This is in line with the present indications for joint replacement, primarily pain and loss of function. Preoperatively, patients are generally able to modify their lives to varying degrees in order to accommodate their arthritis. Therefore, it is reasonable to expect that the patients and their spouses in this study found the greatest benefit of the hip replacement was related to the pain reduction and increased mobility after the total hip replacement. This differs from more life threatening or chronic disease where life modification is drastic or not possible. In these cases, mood/well-being can be significantly affected as well.

Furthermore, I recommend the following paper : PMID 27939622.  It informs well about the subjective view of OA on the patients´ part (and doctors´, too).

Thank you for bringing this article to my attention. After reviewing the article mentioned, it is clear that many of the barriers discussed are dealt with preoperatively in our patients. Although it outside the scope of this study, all patients followed a Clinical Pathway. The procedure was thoroughly, recovery and outcomes were explained by the surgeon, they all met with the nurses and physiotherapists in the preop clinic to discuss the postoperative course and outcomes, and they all received a printed clinical pathway booklet discussing all this. 

That aside, I was wondering about the inclusion criteria: is the study sample representative of a normal clinical sample? Were they screened for any psychiatric disorder? What role did socioeconomic aspects play?

 This is a very important issue that was not addressed in our study. Although we intentionally did not investigate couples that were just married or were divorced, our inclusion criteria were not very strict ie. “were married or co-habiting with their spouse for at least 5 years prior to surgery and were between 65 and 85 years of age.” However, as pointed out, we do not know the quality of their relationship. This is certainly a limitation of the study and is now added to the Limitations paragraph in the Discussion section:

“As well, the quality of the marital relationship and their psychological status was not determined so it is not possible to determine whether or not these couples are representative of a normal clinical sample.”

Reviewer 4 Report

This is a study quantifying the improvement of quality of life after joint replacement surgery for patients with hip OA and their spouses. My major concern to this study are follows:

  • The authors include only patients without severe comorbidity und postoperative complications in this study. It will be more interesting to include all patients in this study, and to provide a more comprehensive picture of the quality of life of all patients and their spouses.
  •  
  • I wonder why the authors let the spouses to rate the severity of pain of the patients. Is there any special reason? How reliable is this kind of rating?
  •  
  • It is well known that joint replacement surgery leads to a tremendous improvement of the ability of daily living of patients with severe hip OA. This study just demonstrates what it should be. The relevance of this research is rather limited.

Author Response

Responses to Reviewer 4

I would like to than the reviewers for their expertise and queries. The manuscript has been modified to respond to and/or clarify the issues raised by the reviewers

This is a study quantifying the improvement of quality of life after joint replacement surgery for patients with hip OA and their spouses. My major concern to this study are follows:

The authors include only patients without severe comorbidity und postoperative complications in this study. It will be more interesting to include all patients in this study, and to provide a more comprehensive picture of the quality of life of all patients and their spouses.

We agree that the patient population is very specific. The patients all have a 100 Harris Hip Score to be included in the study. Of course, this may limit the generality of findings of the study. The limitations paragraph in the Discussion section has been modified to respond to all the reviewer’s comments:

The limitations of this study pertain to the survey nature of it. Consequently, it does not shed light on the particular causes for the disparities between the questionnaire responses of the patients and their spouse. Also, as this study is retrospective in nature, and only done after the arthroplasty, it may be limited by recall bias of both the patient and their spouse. In addition, the Harris Hip Score was only used in this study to include/exclude patients. It does not give any information on patient’s satisfaction or health-related quality of life (HRQoL). The inclusion of only patients with a Harris Hip Score of 100 may have excluded patients with severe comorbidities and postoperative complications, which can impact the generalizability of the study findings. As well, the quality of the marital relationship and their psychological status was not determined so it is not possible to determine whether or not these couples are representative of a normal clinical sample. Finally, the relatively small number of couples in this study may limit the conclusions of this study, as well as the generalizability of these findings to couples of differing socioeconomic status, ethnicity and duration of marriage.

I wonder why the authors let the spouses to rate the severity of pain of the patients. Is there any special reason? How reliable is this kind of rating?

We had the spouses rate the patient’s severity only to document their impression of how much the patient was suffering from their arthritis. It was to determine if the partner perceived that the patient was in fact suffering, and then to determine how it affected their lives. As the reviewer points out, this is purely a subjective finding.

It is well known that joint replacement surgery leads to a tremendous improvement of the ability of daily living of patients with severe hip OA. This study just demonstrates what it should be. The relevance of this research is rather limited.

There is no doubt that the operation has a tremendous improvement on pain and function of the patient. However, there is no literature on the impact of the quality of life of the spouse of the patient. It is not uncommon for a spouse to be urging or asking for their husband/wife to go ahead with surgery when they are reluctant. Instead of discounting the spouse’s pressure to have their husband/wife agree to have surgery, the surgeon should recognize the importance of the surgery for their marital health. Just another indication to consider. Although as surgeons we may tend to underestimate the importance of this, the findings of the study have been picked up by numerous online services, including ABC news, FOX news, AAOS etc since the public feels this is an important finding.

Round 2

Reviewer 1 Report

well done